# Transcatheter aortic valve implantation versus conservative management for severe aortic stenosis in real clinical practice

Yasuaki Takeji[1], Tomohiko Taniguchi[2], Takeshi Morimoto[3], Naritatsu Saito[1]*,
Kenji Ando[2], Shinichi Shirai[2], Genichi Sakaguchi[4], Yoshio Arai[4], Yasushi Fuku[5],
Yuichi Kawase[5], Tatsuhiko Komiya[6], Natsuhiko Ehara[7], Takeshi Kitai[7], Tadaaki Koyama[8],
Shin Watanabe[1], Hirotoshi Watanabe[1], Hiroki Shiomi[1], Eri Minamino-Muta[1],
Shintaro Matsuda[1], Hidenori Yaku[1], Yusuke Yoshikawa[1], Kazuhiro Yamazaki[9],
Masahide Kawatou[9], Kazuhisa Sakamoto[9], Toshihiro Tamura[10], Makoto Miyake[10],
Hisashi Sakaguchi[11], Koichiro Murata[12], Masanao Nakai[13], Norio Kanamori[14],
Chisato Izumi[15], Hirokazu Mitsuoka[16], Masashi Kato[17], Yutaka Hirano[18], Tsukasa Inada[19],
Kazuya Nagao[19], Hiroshi Mabuchi[20], Yasuyo Takeuchi[21], Keiichiro Yamane[22],
Takashi Tamura[23], Mamoru Toyofuku[23], Mitsuru Ishii[24], Moriaki Inoko[25],
Tomoyuki Ikeda[26], Katsuhisa Ishii[27], Kozo Hotta[28], Toshikazu Jinnai[29],
Nobuya Higashitani[29], Yoshihiro Kato[30], Yasutaka Inuzuka[31], Yuko Morikami[32],
Kenji Minatoya[10], Takeshi Kimura[1], on behalf of the CURRENT AS registry Investigators
and the K-TAVI registry Investigators[¶]

1 Department of Cardiovascular Medicine, Graduate School of Medicine, Kyoto University, Kyoto, Japan,
2 Division of Cardiology, Kokura Memorial Hospital, Kitakyushu, Japan, 3 Department of Clinical
Epidemiology, Hyogo College of Medicine, Nishinomiya, Japan, 4 Division of Cardiovascular Surgery, Kokura
Memorial Hospital, Kitakyushu, Japan, 5 Department of Cardiology, Kurashiki Central Hospital, Kurashiki,
Japan, 6 Cardiovascular Surgery, Kurashiki Central Hospital, Kurashiki, Japan, 7 Department of
Cardiovascular Medicine, Kobe City Medical Center General Hospital, Kobe, Japan, 8 Cardiovascular
Surgery, Kobe City Medical Center General Hospital, Kobe, Japan, 9 Department of Cardiovascular Surgery,
Graduate School of Medicine, Kyoto University, Kyoto, Japan, 10 Department of Cardiology, Tenri Hospital,
Tenri, Japan, 11 Cardiovascular Surgery, Tenri Hospital, Tenri, Japan, 12 Department of Cardiology,
Shizuoka City Shizuoka Hospital, Shizuoka, Japan, 13 Cardiovascular Surgery, Shizuoka City Shizuoka
Hospital, Shizuoka, Japan, 14 Division of Cardiology, Shimada Municipal Hospital, Shimada, Japan,
15 Department of Cardiovascular Medicine, National Cerebral and Cardiovascular Center, Suita, Japan,
16 Division of Cardiology, Nara Hospital, Kinki University Faculty of Medicine, Ikoma, Japan, 17 Department
of Cardiology, Mitsubishi Kyoto Hospital, Kyoto, Japan, 18 Department of Cardiology, Kinki University
Hospital, Osakasayama, Japan, 19 Department of Cardiovascular Center, Osaka Red Cross Hospital,
Osaka, Japan, 20 Department of Cardiology, Koto Memorial Hospital, Higashiomi, Japan, 21 Department of
Cardiology, Shizuoka General Hospital, Shizuoka, Japan, 22 Department of Cardiology, Nishikobe Medical
Center, Kobe, Japan, 23 Department of Cardiology, Japanese Red Cross Wakayama Medical Center,
Wakayama, Japan, 24 Department of Cardiology, National Hospital Organization Kyoto Medical Center,
Kyoto, Japan, 25 Cardiovascular Center, The Tazuke Kofukai Medical Research Institute, Kitano Hospital,
Osaka, Japan, 26 Department of Cardiology, Hikone Municipal Hospital, Hikone, Japan, 27 Department of
Cardiology, Kansai Electric Power Hospital, Osaka, Japan, 28 Department of Cardiology, Hyogo Prefectural
Amagasaki General Medical Center, Amagasaki, Japan, 29 Department of Cardiology, Japanese Red Cross
Otsu Hospital, Otsu, Japan, 30 Department of Cardiology, Saiseikai Noe Hospital, Osaka, Japan,
31 Department of Cardiology, Shiga Medical Center for Adults, Moriyama, Japan, 32 Department of
Cardiology, Hirakata Kohsai Hospital, Hirakata, Japan

¶ The complete membership list of CURRENT AS registry Investigators and the K-TAVI registry Investigators
can be found in the Acknowledgments.

* naritatu@kuhp.kyoto-u.ac.jp



BELGIUM

**Data Availability Statement:** All relevant data are
within the paper and its Supporting Information
files.

**Funding:** The CURRENT AS registry was supported by an educational grant from the Research Institute for Production Development (Kyoto, Japan). The funders had no role in study design, data collection and analysis, decision to publish, or preparation of the manuscript. There was no additional external funding received for this study.

**Competing interests:** The authors have declared that no competing interests exist.

# Abstract

## Background

Transcatheter aortic valve implantation (TAVI) is criticized by some as an expensive treatment in super-elder patients with limited life expectancy. However, there is a knowledge gap regarding the magnitude of clinical benefit provided by TAVI in comparison with conservative management in patients with severe aortic stenosis (AS) in real clinical practice, which would be important in the decision making for TAVI.

## Methods

We combined two independent registries, namely CURRENT AS and K-TAVI registries. CURRENT AS was a multicenter registry enrolling 3815 consecutive patients with severe AS irrespective to treatment modalities between January 2003 and December 2011. K-TAVI was a multicenter, prospective registry including 449 consecutive patients with severe AS, who underwent TAVI with SAPIEN XT balloon-expandable valves between October 2013 and June 2016. In these 2 registries, 449 patients received TAVI and 894 patients were managed with conservative strategy. We conducted propensity score matching and finally obtained a cohort of 556 patients (278 patients for each group) for the analysis. The primary outcome measures were all-cause death and heart failure (HF) hospitalization at 2-year.

## Results

The cumulative 2-year incidences of all-cause death and HF hospitalization were significantly lower in the TAVI group than in the conservative group (16.8% versus 36.6%, P<0.001, and 10.7% versus 37.2%, P<0.001). After adjusting the residual confounders, TAVI reduced the risks of all-cause death (HR, 0.46; 95%CI, 0.32–0.69; P = 0.0001) and HF hospitalizations (HR, 0.25; 95%CI, 0.16–0.40; P<0.0001) compared with conservative strategy. There was no difference in the cumulative incidence of non-cardiovascular death between the 2 groups.

## Conclusions

TAVI in the early Japanese experience was associated with striking risk reduction for all-cause death as well as HF hospitalization as compared with the historical cohort of patients with severe AS who were managed conservatively just before introduction of TAVI in Japan.

## Introduction

In symptomatic patients with severe aortic stenosis (AS), surgical aortic valve replacement (SAVR) had been the only option to improve the clinical outcomes, and has been recommended as a class I indication in the guidelines [1–6]. However, one of the biggest drawbacks in the management of patients with severe AS was that substantial proportion of symptomatic patients with severe AS did not receive SAVR due to advanced age, severe comorbidities, or patient rejection [7–9]. Transcatheter aortic valve implantation (TAVI) has already transformed the treatment paradigm of symptomatic patients with severe AS. In severe AS patients

with high or intermediate risk for SAVR, several randomized trials clearly demonstrated that TAVI was associated with the long-term clinical outcomes at least comparable to SAVR [10–14]. Furthermore, in patients with severe AS who were not suitable for SAVR, the PARTNER (Placement of Aortic Transcatheter Valves) trial comparing TAVI with standard treatment demonstrated better outcomes for TAVI up to 5-year follow-up [15–18]. Based on these landmark clinical trials, the proportion of symptomatic severe AS patients treated with aortic valve replacement by either SAVR or TAVI clearly increased after introduction of TAVI [19].

However, there is a knowledge gap regarding how much clinical benefit could be provided by TAVI in comparison with conservative management in patients with severe AS in real clinical practice. The expected magnitude of clinical benefit would be important in the decision making for TAVI in real world patients with severe AS. Against this background, we sought to evaluate the clinical outcomes of patients who underwent TAVI in the early Japanese experience in comparison with the historical cohort of patients who were managed conservatively just before introduction of TAVI in Japan.

## Materials and methods

### Study population

We combined two independent registries in Japan, K-TAVI (Kyoto University-related hospital Transcatheter Aortic Valve Implantation) registry and CURRENT AS (Contemporary outcomes after sURgery and medical tREatmeNT in patients with severe Aortic Stenosis) registry, to make a historical comparison of the clinical outcomes between TAVI and conservative management in patients with severe AS.

K-TAVI registry was a multicenter and prospective registry enrolling consecutive patients with severe AS who underwent TAVI at 6 centers starting from October 2013. The selection of patients and the procedures of the K-TAVI registry were previously reported [20]. For the present analysis, we included 449 patients who underwent TAVI with SAPIEN XT (Edwards Lifesciences, CA, USA) from October 2013 to June 2016 in the K-TAVI registry (Fig 1).

The CURRENT AS registry was a multicenter, retrospective registry that enrolled consecutive patients with severe AS irrespective to treatment modalities from 27 centers (on-site surgical facilities: 20 centers) just before introduction of TAVI in Japan from January 2003 to December 2011. All the 6 centers that participated in the K-TAVI registry had also participated in the CURRENT AS registry. Severe AS was defined as peak aortic jet velocity ($V_{max}$) >4.0m/s, mean aortic pressure gradient (PG) >40mmHg, or aortic valve area (AVA) <1.0cm$^2$. The detailed design and results of the registry have been previously published [21]. Among 3815 patients enrolled in the CURRENT AS registry, conservative management was initially chosen in 2618 patients. To identify the patients with conservative management comparable to the patients in the K-TAVI registry, we excluded those patients on hemodialysis (HD) in whom TAVI has not been yet approved in Japan, and those asymptomatic patients with Vmax <5m/s and left ventricular ejection fraction (LVEF) > = 50%, who are regarded as candidates for watchful waiting according to the guidelines [4]. We also excluded those patients who were regarded as contraindicated for SAVR by the attending physicians (malnutrition, muscle weakness, cognitive impairment, and expected poor prognosis), because these patients were considered to be contraindicated for TAVI. Finally, we retrieved the data of 984 non-HD patients in the conservative group who were symptomatic or asymptomatic but with $V_{max}$ ≥5 m/s, or with LVEF of <50% (Fig 1).

The follow-up was commenced on the day of TAVI in the K-TAVI registry and on the day of index echocardiography in the conservative group from CURRENT AS registry. Follow-up was censored at 2-year in both groups considering the minimal follow-up interval in the K-TAVI registry. We obtained clinical follow-up data from the medical records and/or

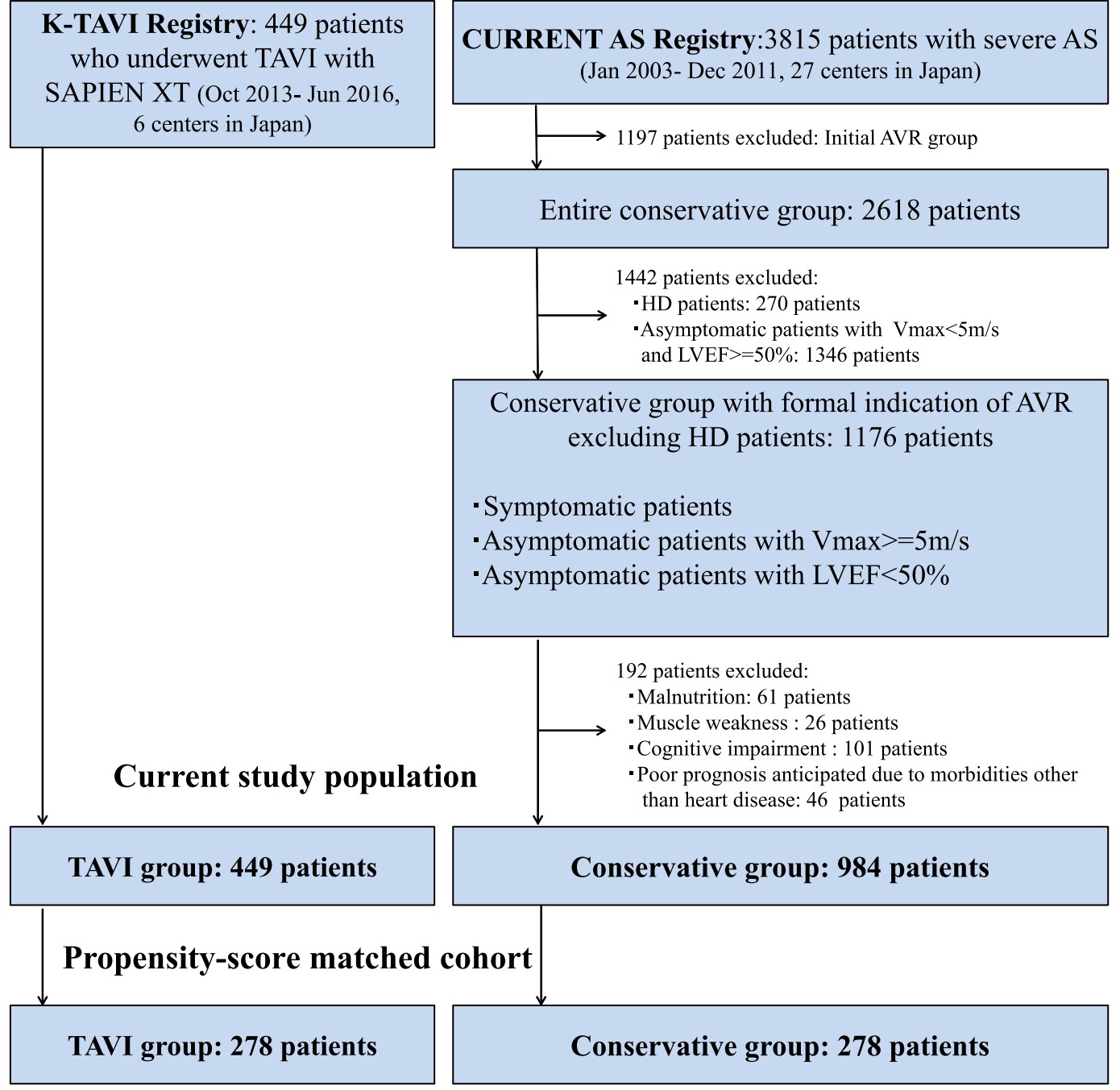

**Fig 1. Study flowchart.** CURRENT AS, Contemporary outcomes after sURgery and medical tREatmeNT in patients with severe Aortic Stenosis; K-TAVI, Kyoto University-related hospital Transcatheter Aortic Valve Implantation; AVR, aortic valve replacement; HD, hemodialysis; Vmax, peak aortic jet velocity; TAVI, transcatheter aortic valve implantation; LVEF, left ventricular ejection fraction.

through mail exchanges and/or telephone interviews with the patients, families, or referring physicians.

The relevant institutional review boards at all participating hospitals approved the study protocols and were described in S3 Text. We performed the study in accordance with the

Declaration of Helsinki. Written informed consent specific for the K-TAVI registry was waived because patients undergoing TAVI provided written informed consent for the compulsory national clinical database registry, and it was also waived in the CURRENT AS registry because of the retrospective study design.

## Study outcomes

Valve implantation was regarded as successful, if the procedure was completed without valve delivery failure, second valve implantation, annulus rupture and conversion to open heart surgery. Device success and other procedural endpoints of TAVI was defined based on the Valve Academic Research Consortium (VARC)-2 classification [22]. The primary outcome measures of the current study were all-cause death and heart failure (HF) hospitalizations at 2-year. The secondary outcome measures included aortic valve-related death, aortic valve procedure death, cardiovascular death, sudden death, non-cardiovascular death, myocardial infarction, stroke, major bleeding, infectious endocarditis, and a composite of aortic valve-related deaths or HF hospitalization. Aortic valve-related death included aortic valve procedure death, sudden death, and death due to HF possibly related to aortic valve. HF hospitalization was defined as hospitalization due to worsening HF requiring intravenous drug therapy. Major bleeding in this study was defined as life-threatening/disabling or major bleeding in the VARC-2 classification. Definitions of other clinical events are described in S4 Text. Clinical events were adjudicated by the clinical event committee (S1 Text) in the CURRENT AS registry, while site-reported events were not adjudicated in the K-TAVI registry.

## Statistical analysis

We expressed continuous variables as mean ± standard deviation or median with interquartile range (IQR), and compared them using Student's t-test or Wilcoxon rank sum test. We expressed categorical variables as percentages and compared them using $\chi^2$ tests.

We used propensity score matching as the main analysis, because the patient characteristics were different between the TAVI and conservative groups derived from the 2 separate registries. Once we combined data of 984 patients from CURRENT AS registry and 449 patients from K-TAVI registry, we used multivariable logistic regression model to develop propensity-score for the choice of TAVI with 13 variables relevant to the choice of AVR used in our previous study (Table 1) [21]. We multiplied these variables in each patient by the coefficients in the model to calculate propensity score of each patient. The c-statistics was 0.818 and the coefficients of the independent variables were shown in S1 Table. We then calculated the propensity score by summing up all coefficients multiplies corresponding variables (S1 Fig). To make propensity-score matched cohort, patients in the TAVI group were matched to those in the conservative group using a 1:1 greedy matching technique [23]. We eliminated those patients without counterparts with corresponding propensity score, and finally constructed the propensity score-matched cohort of 556 patients (TAVI group 278 patients, and conservative group 278 patients), and used Kaplan-Meier curves to estimate cumulative incidences. Log-rank test was used to assess the differences between groups. Because some variables were not well balanced even after the propensity score matching, we performed further adjustment by using the Cox proportional hazard models incorporating the risk-adjusting variables such as Society of Thoracic Surgeons (STS)-predicted risk of mortality (PROM), $V_{max}$, and aortic valve area (AVA). We evaluated hazard ratios (HRs) and their 95% confidence intervals (CIs) to assess the risk of the TAVI group relative to the conservative group for each outcome measure.

As a sensitivity analysis, we constructed Cox proportional hazards models incorporating 18 clinically relevant risk-adjusting variables listed in Table 1 among the entire cohort of 1433

**Table 1. Baseline patient characteristics.**

| | Entire cohort | | | Propensity score-matched cohort | | |
|---|---|---|---|---|---|---|
| | TAVI group | Conservative group | P-value | TAVI group | Conservative group | P-value |
| | (N = 449) | (N = 984) | | (N = 278) | (N = 278) | |
| **Clinical characteristics** | | | | | | |
| Age, *y | 85.2±5.3 | 82.1±9.1 | <0.0001 | 84.6±5.7 | 85.1±7.0 | 0.41 |
| ≥80 years† | 399 (89) | 644 (65) | <0.0001 | 233 (84) | 235 (85) | 0.82 |
| Men*'† | 161 (36) | 288 (29) | 0.01 | 80 (29) | 81 (29) | 0.93 |
| BMI, kg/m$^2$ | 22.0±3.5 | 21.2±3.9 | 0.0001 | 21.4±3.5 | 21.8±3.9 | 0.24 |
| <22.0 kg/m$^{2*}$'† | 235 (52) | 689 (70) | <0.0001 | 171(62) | 171 (62) | 1.00 |
| BSA, m$^2$ | 1.43±0.2 | 1.41±0.2 | 0.07 | 1.40±0.2 | 1.41±0.2 | 0.72 |
| Hypertension* | 349 (78) | 717 (73) | 0.05 | 211 (76) | 211 (76) | 1.00 |
| Smoking* | 82 (18) | 173 (18) | 0.75 | 46 (17) | 47 (17) | 0.91 |
| Dyslipidemia | 220 (49) | 317 (32) | <0.0001 | 140 (50) | 105 (38) | 0.003 |
| Diabetes mellitus | 123 (27) | 206 (21) | 0.008 | 81 (29) | 61 (22) | 0.05 |
| On insulin therapy* | 15 (3.3) | 37 (3.8) | 0.69 | 10 (3.6) | 9 (3.2) | 0.82 |
| Prior MI* | 18 (4.0) | 129 (13) | <0.0001 | 13 (4.7) | 49 (18) | <0.0001 |
| Prior PCI | 126 (28) | 128 (13) | <0.0001 | 75 (27) | 46 (17) | 0.003 |
| Prior CABG | 49 (11) | 71 (7.2) | 0.02 | 27 (9.7) | 33 (12) | 0.41 |
| Prior heart surgery† | 88 (20) | 106 (11) | <0.0001 | 45 (16) | 41 (15) | 0.64 |
| Prior symptomatic stroke*'† | 55 (12) | 131 (13) | 0.58 | 30 (11) | 24 (8.6) | 0.39 |
| Atrial fibrillation or flutter* | 48 (11) | 257 (26) | <0.0001 | 31 (11) | 55 (20) | 0.005 |
| Aortic/peripheral vascular disease* | 71 (16) | 130 (13) | 0.19 | 45 (16) | 35 (13) | 0.23 |
| Serum creatinine, mg/dL* | 0.9 (0.7–1.2) | 0.9 (0.7–1.3) | 0.90 | 0.9 (0.7–1.2) | 1.0 (0.7–1.3) | 0.32 |
| >2mg/dL† | 15 (3.4) | 81 (8.2) | 0.0003 | 12 (4.3) | 11 (4.0) | 0.82 |
| Anemia*'† | 344 (77) | 621 (63) | <0.0001 | 200 (72) | 206 (74) | 0.61 |
| Malignancy*'† | 41 (9.1) | 114 (12) | 0.16 | 23 (8.3) | 19 (6.8) | 0.52 |
| Immunosuppressive therapy† | 21 (4.7) | 36 (3.7) | 0.37 | 11 (4.0) | 10 (3.6) | 0.82 |
| Chronic lung disease | 138 (31) | 109 (11) | <0.0001 | 72 (26) | 29 (10) | <0.0001 |
| moderate or severe*'† | 51 (11) | 44 (4.5) | <0.0001 | 16 (5.8) | 14 (5.0) | 0.71 |
| Coronary artery disease* | 194 (43) | 295 (30) | <0.0001 | 114 (41) | 100 (36) | 0.22 |
| STS score (PROM), % | 6.4 (4.5–9.3) | 5.1 (3.1–8.6) | <0.0001 | 6.4 (4.5–9.2) | 5.8 (4.0–9.5) | 0.13 |
| **Etiology of aortic stenosis** | | | | | | |
| Degenerative | 445 (99) | 916 (93) | <0.0001 | 275 (99) | 268 (96) | 0.22 |
| Congenital (unicuspid, bicuspid, or quadricuspid) | 2 (0.5) | 21 (2.1) | | 1 (0.4) | 3 (1.1) | |
| Rheumatic | 1 (0.2) | 43 (4.4) | | 1 (0.4) | 5 (1.8) | |
| Infective endocarditis | 0 (0) | 0 (0) | | 0 (0) | 0 (0) | |
| Other | 1 (0.2) | 4 (0.4) | | 1 (0.4) | 2 (0.7) | |
| **Echocardiographic variables** | | | | | | |
| V$_{max}$, m/s | 4.7±0.7 | 4.1±1.0 | <0.0001 | 4.6±0.7 | 4.1±1.0 | <0.0001 |
| V$_{max}$ ≥5 m/s† | 154 (34)† | 21 (21) | <0.0001 | 80 (29) | 67 (24) | 0.23 |
| V$_{max}$ ≥4 m/s* | 385 (86) | 526 (53) | <0.0001 | 236 (85) | 151 (54) | <0.0001 |
| Peak aortic PG, mmHg | 87±28 | 70±33 | <0.0001 | 84±26 | 72±35 | <0.0001 |
| Mean aortic PG, mmHg | 52±17 | 40±21 | <0.0001 | 51±17 | 42±22 | <0.0001 |
| AVA, cm$^2$ | 0.62± 0.17 | 0.70±0.19 | <0.0001 | 0.62±0.18 | 0.68±0.19 | <0.0001 |
| AVA index, cm$^2$/m$^2$ | 0.44±0.12 | 0.51±0.14 | <0.0001 | 0.44±0.13 | 0.49±0.14 | <0.0001 |
| Eligibility for severe AS | | | | | | |
| V$_{max}$ >4 m/s or mean aortic PG >40 mmHg | 370 (82) | 505 (51) | <0.0001 | 225 (81) | 147 (53) | <0.0001 |

*(Continued)*

**Table 1.** (Continued)

| | Entire cohort | | | Propensity score-matched cohort | | |
|---|---|---|---|---|---|---|
| | TAVI group | Conservative group | P-value | TAVI group | Conservative group | P-value |
| | (N = 449) | (N = 984) | | (N = 278) | (N = 278) | |
| AVA <1.0 cm$^2$ alone with LVEF <50% | 20 (4.5) | 186 (19) | <0.0001 | 16 (5.8) | 42 (15) | 0.0002 |
| AVA <1.0 cm$^2$ alone with LVEF ≥50% | 56 (12) | 283 (29) | <0.0001 | 36 (13) | 86 (31) | <0.0001 |
| LVDd, mm | 44±7 | 46 ± 7 | <0.0001 | 44±7 | 44 ± 7 | 0.99 |
| LVDs, mm | 29±6 | 31 ± 8 | <0.0001 | 29±7 | 30 ± 7 | 0.56 |
| LVEF, %* | 61±11 | 59 ± 15 | 0.005 | 60±12 | 61 ± 13 | 0.59 |
| <40%† | 21 (4.7) | 123 (13) | <0.0001 | 16 (5.8) | 14 (5.0) | 0.70 |
| <50% | 66 (15) | 259 (26) | <0.0001 | 50 (18) | 59 (21) | 0.35 |
| IVST in diastole, mm | 11±2 | 11 ± 2 | 0.13 | 11 ± 2 | 11 ± 2 | 0.81 |
| PWT in diastole, mm | 11±2 | 11 ± 2 | 0.46 | 11 ± 3 | 11 ± 2 | 0.55 |
| Any combined valvular disease (moderate or severe)*'† | 81 (18) | 509 (52) | <0.0001 | 71 (26) | 70 (25) | 0.92 |
| AR | 33 (7.4) | 234 (24) | <0.0001 | 29 (10) | 31 (11) | 0.80 |
| MS | 16 (3.6) | 39 (4.0) | 0.72 | 5 (1.8) | 14 (5.1) | 0.03 |
| MR | 33 (7.4) | 285 (29) | <0.0001 | 30 (11) | 43 (15) | 0.11 |
| TR | 27 (6.0) | 223 (23) | 0.0001 | 24 (8.7) | 34 (12) | 0.17 |

Categorical variables were presented as number (%), and continuous variables were presented as mean ± SD, or median with interquartile range.

*Potential independent variables selected for Cox proportional hazards models in the unmatched cohort

† Potential independent variables selected for logistic regression model to develop propensity score for the choice of TAVI.

Anemia was defined as serum hemoglobin <12g/dl for women or <13g/dl for men.

TAVI, transcatheter aortic valve implantation; BMI, body mass index; BSA, body surface area; MI, myocardial infarction; PCI percutaneous coronary intervention; CABG, coronary artery bypass grafting; HD, hemodialysis; STS, society of thoracic surgeons; PROM, predicted risk of mortality; V$_{max}$, peak aortic jet velocity; PG, pressure gradient; AVA, aortic valve area; AS, aortic stenosis; LVDd, left ventricular end-diastolic diameter; LVDs, left ventricular end-systolic diameter; LVEF, left ventricular ejection fraction; LV, left ventricular; IVST, interventricular septum thickness; PWT, posterior wall thickness; AR, aortic regurgitation; MS, mitral stenosis; MR, mitral regurgitation; TR, tricuspid regurgitation.

patients (TAVI group, 449 patients, and conservative group, 984 patients). We also performed another sensitivity analysis in the propensity score-matched cohort excluding those patients who died within 30 days after the index echocardiography in the conservative group, because enrollment date of K-TAVI registry was not the index echocardiography date but the TAVI procedure date, and there was possibility that some patients scheduled for TAVI had died before actually undergoing TAVI procedure.

We also performed subgroup analyses in terms of age, sex, STS score, LVEF, and high/low gradient AS in the propensity-score matched cohort. Age and STS score were dichotomized by the median values, while LVEF was dichotomized by > = 50% and <50%.

We considered a 2-sided P-value of <0.05 to be significant for all tests. All analyses were performed using JMP 14.0.0 or SAS 9.4 software (SAS Institute, Cary, NC, USA).

## Results

### Patient characteristics

In the entire cohort, patients in the TAVI group were older than those in the conservative group (Table 1). The age distribution in the range of > = 85 years of age was comparable in

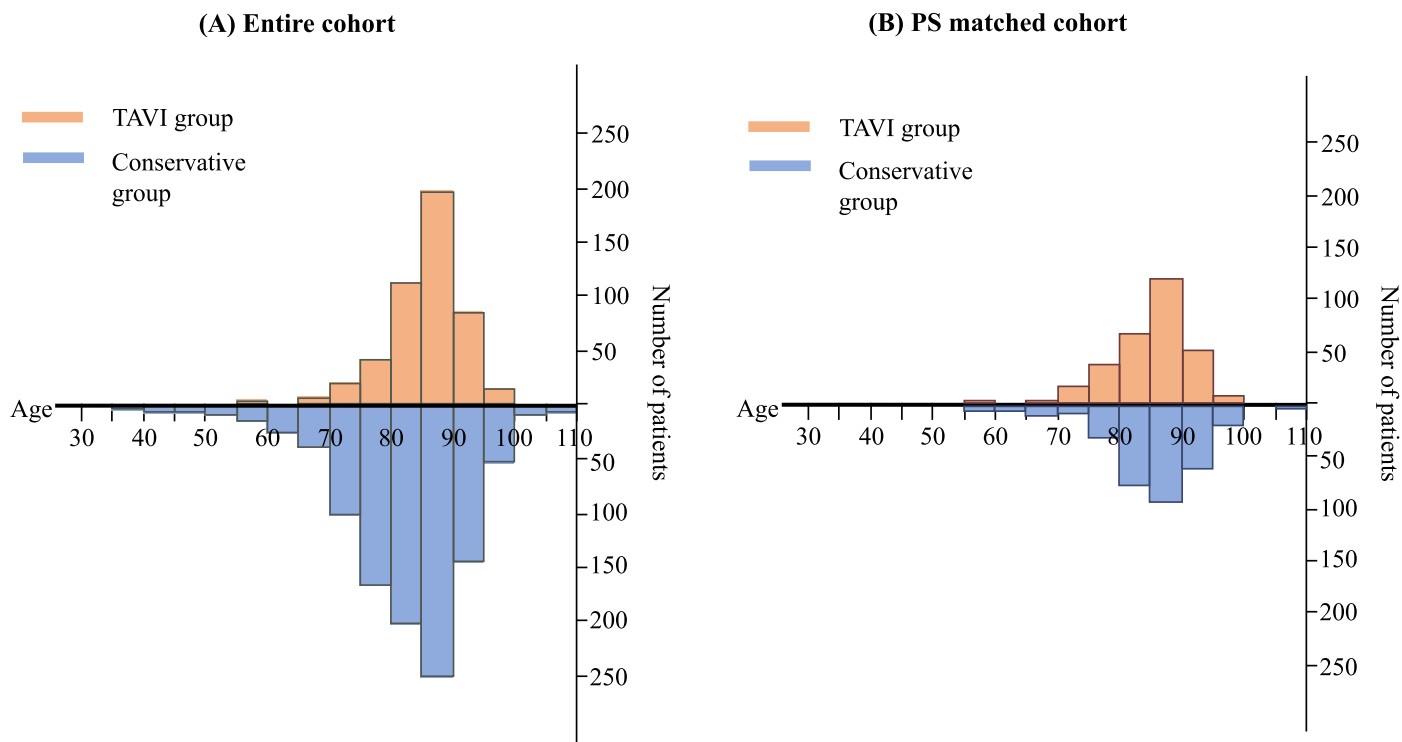

**Fig 2. Distribution of age.** (A) Entire cohort. (B) PS matched cohort (B) PS, propensity score; TAVI, transcatheter aortic valve implantation.

the TAVI group and the conservative group, while the proportion of patients with <85 years of age was smaller in the TAVI group than in the conservative group (Fig 2A). Patients in the TAVI group more often had dyslipidemia, anemia, coronary artery disease, and chronic lung disease and had higher STS score, while patients in the conservative group more often had prior myocardial infarction, atrial fibrillation or flutter, and creatinine levels >2 mg/dL (Table 1). In terms of echocardiographic data, $V_{max}$, mean aortic PG, and LVEF were greater in the TAVI group than in the conservative group. The prevalence of combined valvular disease was much higher in the conservative group than in the TAVI group (Table 1).

In the propensity score-matched cohort, the baseline patient characteristics including STS score were much better balanced between the TAVI and conservative groups (Table 1). Mean age and the age distribution were comparable in the TAVI and conservative groups (Table 1 and Fig 2B). However, patients in the TAVI group still more often had dyslipidemia, prior percutaneous coronary intervention, and chronic lung disease, while patients in the conservative group more often had prior myocardial infarction, and atrial fibrillation or flutter (Table 1). Echocardiographic severity of AS in terms of $V_{max}$, mean PG, and AVA was greater in the TAVI group than in the conservative group even after propensity score matching (Table 1).

## Characteristics and procedural outcomes of TAVI

In terms of procedural characteristics in the TAVI group, trans-femoral approach was selected only in 63% of patients, and the vast majority of patients underwent TAVI under general anesthesia. Successful valve implantation was achieved in 97.3% and device success rate was 92.0% in the entire cohort. The major complications included annulus rupture (0.7%), conversion to

**Table 2. Clinical outcomes: Propensity score-matched cohort.**

| | TAVI group (N = 278) | Conservative group (N = 278) | Hazard Ratio (95% Confidence Interval) | | | |
|---|---|---|---|---|---|---|
| | N of Patients with Event (Cumulative 2-year incidence) | N of Patients with Event (Cumulative 2-year incidence) | Crude | P-value | Adjusted | P-value |
| All-cause death | 45 (16.8%) | 95 (36.6%) | 0.40 (0.28–0.57) | <0.0001 | 0.46 (0.32–0.69) | 0.0001 |
| Cardiovascular death | 21 (8.2%) | 69 (28.0%) | 0.26 (0.16–0.42) | <0.0001 | 0.29 (0.17–0.48) | <0.0001 |
| Aortic valve-related death | 6 (2.3%) | 54 (23.0%) | 0.09 (0.04–0.20) | <0.0001 | 0.10 (0.04–0.22) | <0.0001 |
| Aortic valve procedure death | 5 (1.9%) | 2 (1.0%) | 1.45 (0.36–7.09) | 0.60 | N/A | - |
| Sudden death | 7 (2.8%) | 15 (6.9%) | 0.40 (0.15–0.94) | 0.04 | N/A | - |
| Non-cardiovascular death | 24 (9.4%) | 26 (12.0%) | 0.78 (0.45–1.36) | 0.38 | 1.03 (0.55–1.97) | 0.92 |
| Heart failure hospitalization | 27 (10.7%) | 85 (37.2%) | 0.25 (0.16–0.38) | <0.0001 | 0.25 (0.16–0.40) | <0.0001 |
| Composite of aortic valve-related death or heart failure hospitalization | 32 (12.4%) | 103 (42.4%) | 0.24 (0.16–0.36) | <0.0001 | 0.25 (0.16–0.37) | <0.0001 |
| Myocardial infarction | 1 (0.4%) | 3 (1.5%) | 0.29 (0.01–2.30) | 0.25 | N/A | - |
| Stroke | 12 (4.8%) | 11 (5.3%) | 0.95 (0.42–2.19) | 0.90 | N/A | - |
| Major bleeding | 26 (9.8%) | 13 (5.7%) | 1.88 (0.98–3.78) | 0.06 | N/A | - |
| Infective endocarditis | 6 (2.4%) | 1 (0.5%) | 5.21 (0.89–98.4) | 0.07 | N/A | - |

TAVI, transcatheter aortic valve implantation; N/A, not applicable.

open surgery (0.9%), emergency coronary intervention (0.7%), major vascular complications (4.5%), and permanent pacemaker implantation (4.5%). Median length of hospital stay after TAVI was 12 (IQR: 9–18) days (S2 Table).

## Clinical outcomes in the propensity score-matched cohort

In the propensity score-matched cohort, the cumulative 30-day incidence of all-cause death was significantly lower in the TAVI group than in the conservative group (1.1% and 4.1%, log-rank P = 0.03). The cumulative 30-day incidence of stroke trended to be higher in the TAVI group than in the conservative group (1.8% and 0.4%, log-rank P = 0.11). Cumulative 30-day incidence of major bleeding was significantly higher in the TAVI group than in the conservative group (4.3%, and 0.8%, log-rank P = 0.007) (S3 Table).

For the long-term follow-up in the propensity score-matched cohort, median follow-up intervals of the surviving patients were 809 (IQR: 736–1118) days in the TAVI group and 1155 (IQR: 903–1590) days in the conservative group. During follow-up, 29 patients (10.4%) ultimately underwent SAVR or TAVI in the conservative group.

The cumulative 2-year incidences of the primary outcome measures (all-cause death and HF hospitalization) were significantly lower in the TAVI group than in the conservative group (16.8%, and 36.6%, log-rank P<0.0001, and 10.7% and 37.2%, log-rank P<0.0001) (Table 2, and Fig 3). The cumulative incidences of the secondary outcome measures such as

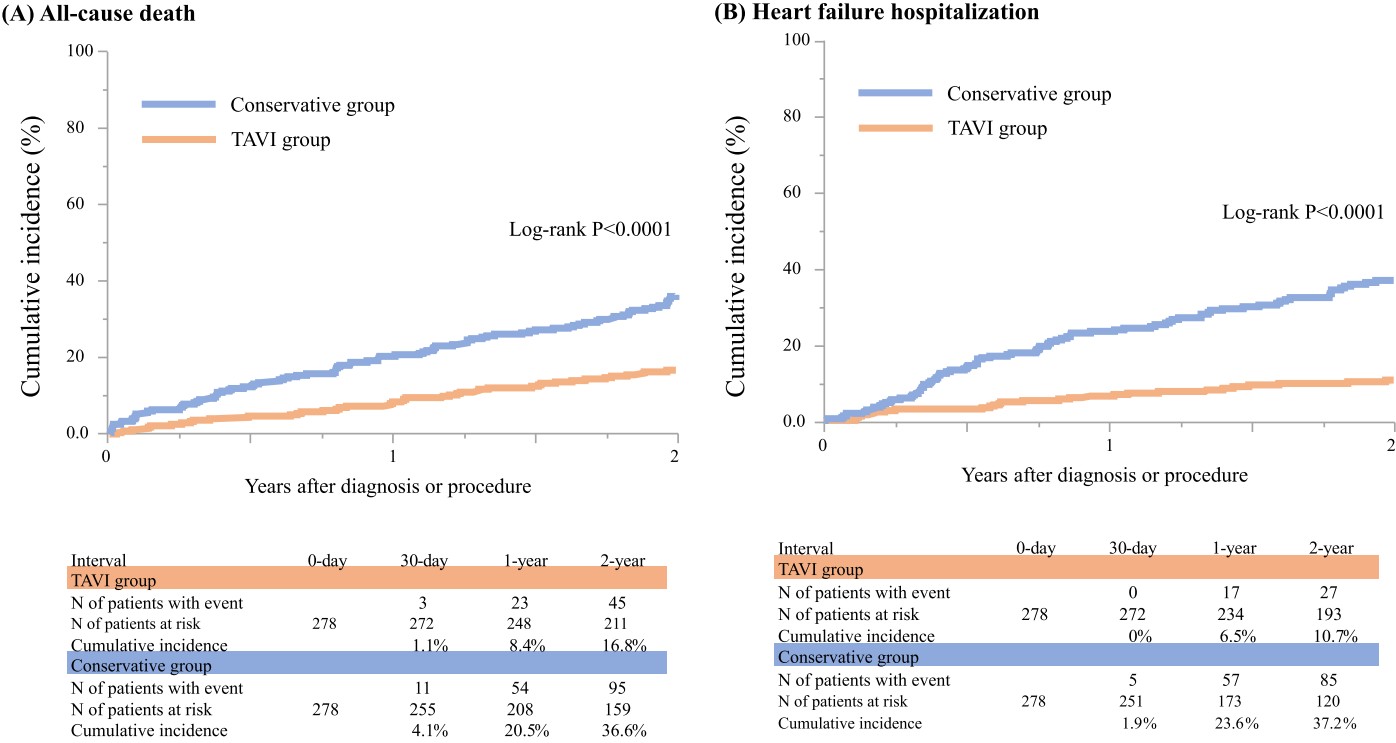

**Fig 3. Kaplan-Meier curves for the primary outcome measures comparing between the TAVI and conservative groups in the PS matched cohort.** (A) all-cause death. (B) heart failure hospitalization.

cardiovascular death, aortic valve-related death, sudden death, and a composite of aortic valve-related death or HF hospitalization were also significantly lower in the TAVI group than in the conservative group (Table 2, and S2 and S3 Figs). The cumulative incidences of non-cardiovascular death, aortic valve procedure death, stroke, and myocardial infarction were not significantly different between the 2 groups (Table 2, and S3 and S4 Figs). The cumulative incidences of major bleeding and infectious endocarditis trended to be higher in the TAVI group than in the conservative group (Table 2, and S4 Fig).

After adjustment for the residual confounders, TAVI as compared with conservative management was associated with highly significant risk reduction for all-cause death and HF hospitalization (HR 0.46, 95%CI, 0.32–0.69, P = 0.0001, and HR 0.25, 95%CI 0.16–0.40, P<0.0001) (Table 2). The magnitude of risk reduction with TAVI relative to conservative management for the aortic valve related outcome measure (a composite of aortic valve-related death or HF hospitalization) was comparable to that for HF hospitalization (HR, 0.25, 95%CI, 0.16–0.37; P<0.0001) (Table 2).

## Sensitivity analyses

In the entire cohort, median follow-up intervals of the surviving patients were 846 (IQR: 736–1127) days in the TAVI group and 1294 (IQR: 980–1701) days in the conservative group. During follow-up, 134 patients (13.6%) ultimately underwent SAVR or TAVI in the conservative group. The adjusted risks of the TAVI group relative to the conservative group for the primary outcome measures in the entire cohort were fully consistent with those in the propensity score-matched cohort (S4 Table, and S5–S8 Figs). The results were also consistent in another

sensitivity analysis in the propensity score-matched cohort excluding those patients who died within 30 days after the index echocardiography in the conservative group (S5 Table).

## Subgroup analyses

In the subgroup analyses, there was no significant interaction between the subgroup factors and the effect of TAVI relative to the conservative management for the primary outcome measures, except for the positive interaction between sex and the effect for all-cause death (S9 Fig).

## Discussion

The main finding of the present study was that TAVI in the early Japanese experience was associated with striking risk reduction for all-cause death as well as HF hospitalization as compared with the historical cohort of patients with severe AS who were managed conservatively just before introduction of TAVI in Japan.

TAVI is now widely accepted and has already revolutionized the treatment of severe AS. TAVI have been has been adopted rapidly for patients who are at high surgical risk in the world [24–27]. However, TAVI is criticized by some as an expensive treatment in super-elder patients with limited life expectancy. Measuring the magnitude of benefit provided by TAVI as compared with conservative management is essential to discuss the cost-effectiveness of TAVI. In the PARTNER randomized trial, TAVI as compared with standard treatment was associated with relative 44% risk reduction for all-cause death and 59% risk reduction for re-hospitalization at 2-year follow-up in patients with severe AS who were not suitable for SAVR [16]. In the real clinical practice, however, conservative management had often been selected in symptomatic severe AS patients who are high-risk but not unsuitable for SAVR, while TAVI has often been chosen in this group of patients. Therefore, the magnitude of benefit provided by TAVI as compared with conservative management could not be fully assessed in the PART-NER randomized trial. However, there was no previous study exploring how much clinical benefit could be provided by TAVI in comparison with conservative management in patients with severe AS in the real clinical practice. The 2 registries analyzed in the present study, one in the pre-TAVI era, and the other in the TAVI era, could present a unique opportunity to assess the clinical impact of TAVI relative to conservative management in the real world patients with severe AS. In the present propensity score matched analysis, TAVI as compared with conservative management was associated with striking 54% risk reduction for all-cause death and 75% risk reduction for HF hospitalization at 2-year follow-up. High rate of successful valve implantation, low complication rate and very low 30-day mortality rate were also striking, given the fact that this was the very early TAI experience using the prototype device in Japan. Initial procedural risk is usually the tax to pay for the expected long-term benefit of any invasive treatment. However, in the present study, 30-day mortality was significantly lower in the TAVI group than in the conservative group, highlighting the low procedural risk of TAVI as well as very poor prognosis of patients with severe AS when managed conservatively. Furthermore, the 2-year mortality rate after TAVI was only 16.8% in the present study as compared with 43.3% in the PARTNER trial, indicating that the life expectancy of patients undergoing TAVI in the real world was not so short as shown among the inoperable patients enrolled in the PARTNER trial. In line with the marked reduction of HF hospitalization by TAVI in the present study, TAVI is also reported to be associated with marked improvement of symptoms and quality of life [28–31]. The present study is not a formal cost-effectiveness analysis of TAVI. Nevertheless, given the substantial mortality and morbidity benefit of TAVI, the door to TAVI should not be closed due to the cost issues.

There were some negative aspects for TAVI in the present study. Stroke and major bleeding at 30-day were more frequent in the TAVI group than in the conservative group. The long-term risk for infective endocarditis trended to be higher in the TAVI group than in the conservative group. However, the observed mortality and morbidity benefit of TAVI far outweighed these negative aspects for TAVI. The use of newer generation devices has already reduced the incidence of peri-operative stroke and major bleeding [14,32].

There are several important limitations in this study. First, we combined 2 different registries for the present analysis. We developed the propensity-score for the choice of TAVI in the data set derived from 2 different registries, which might not be a formal way of developing propensity-score. However, the sensitivity analysis using multivariable Cox proportional hazard model in the entire cohort provided the fully consistent results with the propensity-score matched analysis. Nevertheless, we could not exclude the possibility of unmeasured confounding. Second, we conducted a comparison between the 2 registries that enrolled patients just before and after introduction of TAVI in Japan, in which the limitations associated with historical comparison were inevitable. However, we do not have good methodology other than historical comparison to estimate the impact of TAVI in the real clinical practice. The 2 multi-center observational studies conducted among the same group of investigators actually provided very unique opportunity to estimate the magnitude of benefit provided by TAVI. Third, we did not know the number of patients who were turned down for TAVI during enrollment in K-TAVI registry. However, we excluded those patients in the CURENT AS registry who were regarded as contraindicated for SAVR by the attending physicians, because some of these patients might also be contraindicated for TAVI. Fourth, we did not assess the symptomatic status of patients in the K-TAVI registry. History of acute HF hospitalization, which would have substantial prognostic impact, could not be adjusted in the comparison between TAVI and conservative management. Fifth, patients who underwent TAVI were early experience data in Japan, therefore about 37% of patients were selected alternative approach and almost all patients underwent TAVI under general anesthesia. This is quite different from current TAVI practice and this could not applicable to current TAVI practice. Finally, follow-up was commenced at different time points in the 2 registries (TAVI group: the day of TAVI, and conservative group: the day of index echocardiography). Therefore, we conducted a sensitivity analysis excluding those patients who died within 30 days after entry in the conservative group, demonstrating results that are fully consistent with those in the main analysis.

## Conclusions

TAVI in the early Japanese experience was associated with striking risk reduction for all-cause death as well as HF hospitalization as compared with the historical cohort of patients with severe AS who were managed conservatively just before introduction of TAVI in Japan.

## Supporting information

**S1 Text. Study Organization.**
(DOCX)

**S2 Text. List of participating centers and investigators.**
(DOCX)

**S3 Text. List of relevant institutional review boards.**
(DOCX)

**S4 Text. Definitions of the endpoints.**
(DOCX)

**S1 Fig.** Distribution of propensity score in (A) the entire cohort and (B) PS matched cohort. (DOCX)

**S2 Fig.** Kaplan-Meier curves for (A) cardiovascular death and (B) composite of aortic valve-related death or heart failure hospitalization in the PS matched cohort. (DOCX)

**S3 Fig.** Kaplan-Meier curves for (A) aortic valve-related death, (B) aortic valve procedure death, (C) sudden death, and (D) non-cardiovascular death in the PS matched cohort. (DOCX)

**S4 Fig.** Kaplan-Meier curves for (A) myocardial infarction, (B) Stroke, (C) major bleeding and (D) infective endocarditis in the PS matched cohort. (DOCX)

**S5 Fig.** Kaplan-Meier curves for (A) all-cause death and (B) heart failure hospitalization in the entire cohort. (DOCX)

**S6 Fig.** Kaplan-Meier curves for (A) cardiovascular death and (B) composite of aortic valve-related death or heart failure hospitalization in the entire cohort. (DOCX)

**S7 Fig.** Kaplan-Meier curves for (A) aortic valve-related death, (B) aortic valve procedure death, (C) sudden death, and (D) non-cardiovascular death in the entire cohort. (DOCX)

**S8 Fig.** Kaplan-Meier curves for (A) myocardial infarction, (B) Stroke, (C) major bleeding and (D) infective endocarditis in the entire cohort. (DOCX)

**S9 Fig.** Subgroup analysis for the primary outcome measure: (A) All-cause moratality and (B) Heart failure hospitalization. (DOCX)

**S1 Table. Coefficients of the independent variables in the logistic regression function.** (DOCX)

**S2 Table. Procedural characteristics and outcomes of the patients who underwent TAVI.** (DOCX)

**S3 Table. Clinical outcomes at 30-day in the PS-matched cohort and in the entire cohort.** (DOCX)

**S4 Table. Clinical outcomes in the entire cohort.** (DOCX)

**S5 Table. Clinical outcomes in the PS-matched cohort after excluding those patients who died within 30 days after the index echocardiography in the conservative group.** (DOCX)

## Acknowledgments

We deeply appreciate the following co-investigators of the participating centers: Kyoto University; Takeshi Kimura, Naritatsu Saito, Shin Watanabe, Hirotoshi Watanabe, Hiroki Shiomi,

Eri Minamino-Muta, Shintaro Matsuda, Hidonori Yaku, Yusuke Yoshikawa, Yasuaki Takeji, Tomoki Sasa, Masao Imai, Junichi Tazaki, Toshiaki Toyota,

Hirooki Higami, Tetsuma Kawaji, Ryuzo Sakata, Kenji Minatoya, Kenji Minakata, Kazuhiro Yamazaki, Masahide Kawatou, Kazuhisa Sakamoto, Shinya Takimoto, Kokura Memorial Hospital; Kenji Ando, Shinichi Shirai, Tomohiko Taniguchi, Kengo Kourai, Takeshi Arita, Shiro Miura, Michiya Hanyu, Genichi Sakaguchi, Yoshio Arai, Hyogo College of Medicine; Takeshi Morimoto, Kurashiki Central Hospital; Kazushige Kadota, Yasushi Fuku, Yuichi Kawase, Tatsuhiko Komiya, Keiichiro Iwasaki, Hiroshi Miyawaki, Ayumi Misao, Akimune Kuwayama, Masanobu Ohya, Takenobu Shimada, Hidewo Amano, Kobe City Medical Center General Hospital; Yutaka Furukawa, Natsuhiko Ehara, Takeshi Kitai, Tadaaki Koyama, Tenri Hospital; Yoshihisa Nakagawa, Toshihiro Tamura, Makoto Miyake, Masashi Amano, Yusuke Takahashi, Shunsuke Nishimura, Maiko Kuroda, Kazuo Yamanaka, Hisashi Sakaguchi, Shizuoka City Shizuoka Hospital; Tomoya Onodera, Koichiro Murata, Fumio Yamazaki, Masanao Nakai, Shimada Municipal Hospital; Takeshi Aoyama, Norio Kanamori, National Cerebral and Cardiovascular Center; Chisato Izumi, Nara Hospital, Kinki University Faculty of Medicine; Manabu Shirotani, Hirokazu Mitsuoka, Noboru Nishiwaki, Mitsubishi Kyoto Hospital; Shinji Miki, Tetsu Mizoguchi, Masashi Kato, Takafumi Yokomatsu, Akihiro Kushiyama, Toshimitsu Watanabe, Kenji Nakatsuma, Hiroyuki Nakajima, Motoaki Ohnaka, Hiroaki Osada, Katsuaki Meshii, Kinki University Hospital; Shunichi Miyazaki, Yutaka Hirano, Toshihiko Saga, Osaka Red Cross Hospital, Tsukasa Inada, Kazuya Nagao, Naoki Takahashi, Kohei Fukuchi, Shogo Nakayama, Koto Memorial Hospital; Tomoyuki Murakami, Hiroshi Mabuchi, Teruki Takeda, Tomoko Sakaguchi, Keiko Maeda, Masayuki Yamaji, Motoyoshi Maenaka, Yutaka Tadano, Shizuoka General Hospital; Hiroki Sakamoto, Yasuyo Takeuchi, Makoto Motooka, Nishikobe Medical Center; Hiroshi Eizawa, Keiichiro Yamane, Mitsunori Kawato, Minako Kinoshita, Kenji Aida, Japanese Red Cross Wakayama Medical Center; Takashi Tamura, Mamoru Toyofuku, Kousuke Takahashi, Euihong Ko, Atsushi Iwakura, National Hospital Organization Kyoto Medical Center; Masaharu Akao, Mitsuru Ishii, Nobutoyo Masunaga, Hisashi Ogawa, Moritake Iguchi, Takashi Unoki, Kensuke Takabayashi, Yasuhiro Hamatani, Yugo Yamashita, Kotaro Shiraga, Kishiwada City Hospital; Mitsuo Matsuda, Sachiko Sugioka, Masahiko Onoe, Kitano Hospital; Moriaki Inoko, Takao Kato, Koji Ueyama, Hikone Municipal Hospital; Yoshihiro Himura, Tomoyuki Ikeda, Kansai Electric Power Hospital; Katsuhisa Ishii, Akihiro Komasa, Hyogo Prefectural Amagasaki General Medical Center; Yukihito Sato, Kozo Hotta, Shuhei Tsuji, Keiichi Fujiwara, Japanese Red Cross Otsu Hospital; Takashi Konishi, Toshikazu Jinnai, Nobuya Higashitani, Kouji Sogabe, Michiya Tachiiri, Yukiko Matsumura, Chihiro Ota, Mitsuru Kitano, Rakuwakai Otowa Hospital; Yuji Hiraoka, Atsushi Fukumoto, Hamamatsu Rosai Hospital; Eiji Shinoda, Miho Yamada, Akira Kawamoto, Chiyo Maeda, Junichiro Nishizawa, Saiseikai Noe Hospital; Ichiro Kouchi, Yoshihiro Kato, Shiga Medical Center for Adults; Shigeru Ikeguchi, Yasutaka Inuzuka, Soji Nishio, Jyunya Seki, Masaki Park, Hirakata Kohsai Hospital, Shoji Kitaguchi, Yuko Morikami

## Author Contributions

**Conceptualization:** Takeshi Morimoto, Takeshi Kimura.

**Data curation:** Yasuaki Takeji, Tomohiko Taniguchi.

**Formal analysis:** Yasuaki Takeji, Takeshi Morimoto.

**Methodology:** Takeshi Morimoto, Takeshi Kimura.

**Project administration:** Tomohiko Taniguchi, Takeshi Morimoto, Naritatsu Saito, Kenji Ando, Shinichi Shirai, Genichi Sakaguchi, Yoshio Arai, Yasushi Fuku, Yuichi Kawase,

Tatsuhiko Komiya, Natsuhiko Ehara, Takeshi Kitai, Tadaaki Koyama, Shin Watanabe, Hirotoshi Watanabe, Hiroki Shiomi, Eri Minamino-Muta, Shintaro Matsuda, Hidenori Yaku, Yusuke Yoshikawa, Kazuhiro Yamazaki, Masahide Kawatou, Kazuhisa Sakamoto, Toshihiro Tamura, Makoto Miyake, Hisashi Sakaguchi, Koichiro Murata, Masanao Nakai, Norio Kanamori, Chisato Izumi, Hirokazu Mitsuoka, Masashi Kato, Yutaka Hirano, Tsukasa Inada, Kazuya Nagao, Hiroshi Mabuchi, Yasuyo Takeuchi, Keiichiro Yamane, Takashi Tamura, Mamoru Toyofuku, Mitsuru Ishii, Moriaki Inoko, Tomoyuki Ikeda, Katsuhisa Ishii, Kozo Hotta, Toshikazu Jinnai, Nobuya Higashitani, Yoshihiro Kato, Yasutaka Inuzuka, Yuko Morikami, Kenji Minatoya, Takeshi Kimura.

**Software:** Takeshi Morimoto.

**Supervision:** Tomohiko Taniguchi, Takeshi Morimoto, Naritatsu Saito, Takeshi Kimura.

**Validation:** Yasuaki Takeji, Takeshi Morimoto.

**Writing – original draft:** Yasuaki Takeji.

**Writing – review & editing:** Takeshi Morimoto, Naritatsu Saito, Takeshi Kimura.

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
