## [Decision Letter · Decision Letter 0]

1 Aug 2019

PONE-D-19-17543

Transcatheter Aortic Valve Implantation Versus Conservative Management for Severe Aortic Stenosis in Real Clinical Practice

PLOS ONE

Dear Dr Naritatsu Saito,

Thank you for submitting your manuscript to PLOS ONE. After careful consideration, we feel that it has merit but does not fully meet PLOS ONE’s publication criteria as it currently stands. Therefore, we invite you to submit a revised version of the manuscript that addresses the points raised during the review process.

In the revised version of the manuscript, you should provide some lacking informations on patients, including inclusion/exclusion criteria. Some re-analyses should be performed in line with updated thresholds from recent studies. Satistics methods need to be adapted according to reviewer's comments.

We would appreciate receiving your revised manuscript by 31st October. To enhance the reproducibility of your results, we recommend that if applicable you deposit your laboratory protocols in protocols.io, where a protocol can be assigned its own identifier (DOI) such that it can be cited independently in the future. For instructions see: http://journals.plos.org/plosone/s/submission-guidelines#loc-laboratory-protocols

We look forward to receiving your revised manuscript.

Kind regards,

Cécile Oury

Academic Editor

PLOS ONE

Journal Requirements:

2. Please provide details of the author contributions in accordance with CRediT standards https://journals.plos.org/plosone/s/authorship

*In your financial disclosure, please clearly specify whether the funders played any role in the study.

3. Please provide the names of all the ethics committees which approved this study.

*Please include the information on ethics approval and consent, given in the methods section of your manuscript, in the ethics statement on the online submission form.

 [Yes

 CURRENT AS registry was supported by an educational grant from the Research Institute for Production Development (Kyoto, Japan).]. 

Reviewers' comments:

Reviewer's Responses to Questions

**Comments to the Author**

1. Is the manuscript technically sound, and do the data support the conclusions?

Reviewer #1: Yes

Reviewer #2: No

2. Has the statistical analysis been performed appropriately and rigorously? 

Reviewer #1: Yes

Reviewer #2: No

3. Have the authors made all data underlying the findings in their manuscript fully available?

Reviewer #1: Yes

Reviewer #2: Yes

4. Is the manuscript presented in an intelligible fashion and written in standard English?

Reviewer #1: Yes

Reviewer #2: Yes

5. Review Comments to the Author

Reviewer #1: This is a nice study combining two prospective registries. The study is providing new and valuable data.

The references are perhaps missing European registries’data.

The paper is nicely written and very informative, the discussion is really nice and the figures are clean.

Perhaps the text is a bit too long? It could be shorten to provide more data on subgroups with figures.

It appears that the study is not including patients with any low gradient, low flow AS? That is a pity and adding a specific analysis on this group of patients could be even more original and valuable as no strong data are available yet.

The EF is looked according to cut-off of 40 and 50% but the recent literature is underscoring the difference between 60 or 55% versus below: could the authors provide these data and the Kaplan Meier curves according to EF above of below 55% when the patients are treated or not?

The COPD is strikingly different between groups: comments.

The ischemic heart disease: comments?

Is it expected to have such a low rate of AF in such elderly patients? This is strikingly different for USA and Europe.

Reviewer #2: General comments:

In this observational retrospective study authors aimed at comparing TAVI to medical treatment using historical cohort data. The study is not timely since the question asked has been previously answered in much more robust studies. However, it could still be interesting regarding the long-term differences between the two groups. But authors should provide a more robust methodological and statistical approach and would benefit from simplification (the sensitivity and subgroup analyses provide little new insight).

Specific comments:

• Not a timely study, as attested by the use of Sapien XT valves, 37% of alternative access, with probably transapical as first alternative (Results page 22 “trans-femoral approach was selected only in 63% of patients”) with a comparator based on a historical pre-TAVI era cohort. Must be clearly stated that comparison is not applicable to current clinical practice

• Method page 15 “We also excluded those patients … might also be contraindicated for TAVI” Authors should clarify what patients were excluded here and reformulate this statement because TAVI is a clear indication for a large proportion of patients which are contraindicated to SAVR. Please also clarify the flow chart figure 1 with regard to the box “conservative group with formal indication of AVR …”

• Statistical analysis page 17 “Because some variables were not well balanced … further adjustment by using the Cox proportional hazard models” In case of PS matching failure please consider removing PS matching step and apply multivariable Cox upfront.

• Statistical analysis “We also performed subgroup analyses in terms of age, sex, STS score, LVEF, and high/low gradient AS in the propensity-score matched cohort” As reported by authors, the PS matching failed, therefore building subgroups for this cohort is likely to provide biased results.

• Please consider separating all-cause mortality and HR rehospitalization all over the manuscript. Those 2 outcomes have little in common and the database is likely able to provide these data.

• Please provide information regarding TAVI procedural characteristics in a Table: valves used, labels/generations, pathways, TEE use, general versus local anaesthesia, complications, etc

6. PLOS authors have the option to publish the peer review history of their article (what does this mean?). If published, this will include your full peer review and any attached files.

Reviewer #1: No

Reviewer #2: Yes: Thomas Modine, Pavel Overtchouk

---

## [Author Response · Author response to Decision Letter 0]

12 Aug 2019

Reply to the comments of the editors and reviewers

We truly appreciate the time and effort invested by the editors and reviewers in providing us with a critical assessment on our manuscript. We revised our manuscript according to the comments and suggestions of the editors and reviewers.

We showed all of the response to editors and reviewers with reference tables and figures in "Response to reviewer file" added on the end of "revised file".

<Response to the comments of editors>

Thank you for your guidance. We confirmed our script meet PLOS ONE's style requirements.

2. Please provide details of the author contributions in accordance with CRediT standards 

*In your financial disclosure, please clearly specify whether the funders played any role in the study.

Thank you for your guidance. We confirmed and describe the author contributions in accordance with CRediT standards as follows.

Author Contributions (Page 23, Line 16- Page 24, Line 12)

Conceptualization: Takeshi Morimoto, Takeshi Kimura. 

Data curation: Yasuaki Takeji, Tomohiko Taniguchi

Methodology: Takeshi Morimoto, Takeshi Kimura.

Formal analysis: Yasuaki Takeji, Takeshi Morimoto.

Project administration: Tomohiko Taniguchi, Takeshi Morimoto, Naritatsu Saito, Kenji Ando, Shinichi Shirai, Genichi Sakaguchi, Yoshio Arai, Yasushi Fuku, Yuichi Kawase, Tatsuhiko Komiya, Natsuhiko Ehara, Takeshi Kitai, Tadaaki Koyama, Shin Watanabe, Hirotoshi Watanabe, Hiroki Shiomi, Eri Minamino-Muta, Shintaro Matsuda, Hidenori Yaku, Yusuke Yoshikawa, Kazuhiro Yamazaki, Masahide Kawatou, Kazuhisa Sakamoto, Toshihiro Tamura, Makoto Miyake, Hisashi Sakaguchi, Koichiro Murata, Masanao Nakai, Norio Kanamori, Chisato Izumi, Hirokazu Mitsuoka, Masashi Kato, Yutaka Hirano, Tsukasa Inada, Kazuya Nagao, Hiroshi Mabuchi, Yasuyo Takeuchi, Keiichiro Yamane, Takashi Tamura, Mamoru Toyofuku, Mitsuru Ishii, Moriaki Inoko, Tomoyuki Ikeda, Katsuhisa Ishii, Kozo Hotta, Toshikazu Jinnai, Nobuya Higashitani, Yoshihiro Kato, Yasutaka Inuzuka, Yuko Morikami, Kenji Minatoya, Takeshi Kimura.

Software: Takeshi Morimoto.

Supervision: Naritatsu Saito, Tomohiko Taniguchi, Takeshi Morimoto, Takeshi Kimura.

Validation: Yasuaki Takeji, Takeshi Morimoto.

Writing – original draft: Yasuaki Takeji.

Writing – review & editing: Naritatsu Saito, Takeshi Morimoto, Takeshi Kimura

3. Please provide the names of all the ethics committees which approved this study.

*Please include the information on ethics approval and consent, given in the methods section of your manuscript, in the ethics statement on the online submission form.

We appreciate your guidance. 

The following sentence was placed in the Materials and Methods section of our manuscript and The relevant institutional review boards at all participating hospitals were described in S3 Text.

Materials and Methods (Page 8, Line 21- Page 9, Line 2)

The relevant institutional review boards at all participating hospitals approved the study protocols and were described in S3 text. We performed the study in accordance with the Declaration of Helsinki. Written informed consent specific for the K-TAVI registry was waived because patients undergoing TAVI provided written informed consent for the compulsory national clinical database registry, and it was also waived in the CURRENT AS registry because of the retrospective study design.

 [Yes

 CURRENT AS registry was supported by an educational grant from the Research Institute for Production Development (Kyoto, Japan).]. 

We appreciate your guidance. 

We included my amended Funding Statement within your cover letter and manuscript as follows. 

CURRENT AS registry was supported by an educational grant from the Research Institute for Production Development (Kyoto, Japan). The funders had no role in study design, data collection and analysis, decision to publish, or preparation of the manuscript. There was no additional external funding received for this study.

 

<Response to the comments of the reviewer #1>

Reviewer #1:

#1.This is a nice study combining two prospective registries. The study is providing new and valuable data. The references are perhaps missing European registries’data.

The paper is nicely written and very informative, the discussion is really nice and the figures are clean. 

We appreciate your comment. As we mentioned in text, to evaluate the clinical benefit of TAVI compared with conservative therapy have been still important for clinical decision. Not only randomized control trials but also real-world data was important regarding about this problem. 

We added the following description and reference from European registries’ reference in the Discussion section.

Discussion (Page 18, Line 25 to Page 19, Line 1)

TAVI have been has been adopted rapidly for patients who are at high surgical risk in the world [24-27]. 

[24] Gilard M, Eltchaninoff H, Iung B, Donzeau-Gouge P, Chevreul K, Fajadet J, et al. Registry of transcatheter aortic-valve implantation in high-risk patients. The New England journal of medicine. 2012;366(18):1705-15.

[25] Ludman PF, Moat N, de Belder MA, Blackman DJ, Duncan A, Banya W, et al. Transcatheter aortic valve implantation in the United Kingdom: temporal trends, predictors of outcome, and 6-year follow-up: a report from the UK Transcatheter Aortic Valve Implantation (TAVI) Registry, 2007 to 2012. Circulation. 2015;131(13):1181-90.

[26] Krasopoulos G, Falconieri F, Benedetto U, Newton J, Sayeed R, Kharbanda R, et al. European real world trans-catheter aortic valve implantation: systematic review and meta-analysis of European national registries. Journal of cardiothoracic surgery. 2016;11(1):159.

[27] Auffret V, Lefevre T, Van Belle E, Eltchaninoff H, Iung B, Koning R, et al. Temporal Trends in Transcatheter Aortic Valve Replacement in France: FRANCE 2 to FRANCE TAVI. Journal of the American College of Cardiology. 2017;70(1):42-55.

#2.Perhaps the text is a bit too long? It could be shorten to provide more data on subgroups with figures.

Thank you for your comment. In my text, we described introduction, materials and methods, results and discussion. We described details of the materials and methods because of this complex methodology, and both results and discussion had important message in our study.

On the other hand, in subgroup analysis, we only wanted to demonstrate consistency about primary outcomes, and we did not aim to obtain new finding from subgroup analysis.

#3.It appears that the study is not including patients with any low gradient, low flow AS? That is a pity and adding a specific analysis on this group of patients could be even more original and valuable as no strong data are available yet.

Thank you for your valuable comment. In this study, we also included patients with low gradient AS. However, we did not have data of stroke volume , so we could not identify patients with low flow, low gradient severe AS. 

#4.The EF is looked according to cut-off of 40 and 50% but the recent literature is underscoring the difference between 60 or 55% versus below: could the authors provide these data and the Kaplan Meier curves according to EF above of below 55% when the patients are treated or not?.

Thank you very much for your valuable suggestion. Cut-off of 40 and 50% were used in prior our study and we adopted the same value. These days, in the field of heart failure, heart failure with mid- range ejection fraction was hot topic and this was defined as EF 40-49%. We set also heart failure hospitalization as one of primary endpoint in this study. Regarding this point, our group consider that the cut-off point 40 and 50 were reasonable. However, for post-hoc analysis, we conducted Kaplan Meier curves according to EF above or below 55% and cumulative incidence and adjusted hazard ratio divided by EF 55 and 60% in "Response to reviewer file" added on the end of "revised file". 

The results of both cut-off points were consistent with cut-off of 50%.

#5.The COPD is strikingly different between groups: comments..

Thank you for your valuable comments.

As you pointed out, much more patients had chronic lung disease in the TAVI group than in the conservative group. We consider this was because TAVI procedure in this study was at the early experience after introduction of TAVI in Japan. In this era, Japanese physicians selected TAVI when patients have high risk of surgical aortic valve replacement because of comorbidities like chronic lung disease. This is proof that 29.5% of the patients who underwent TAVI had moderate and severe COPD in Japanese National TAVI registry [1]. However, after propensity score matched cohort, the difference of rate of moderate or severe chronic lung disease was set off.

#6.The ischemic heart disease: comments?

Thank you for your valuable comments.

As you pointed out, patients in the TAVI group had more coronary artery disease than in the conservative group. We consider this is because before TAVI procedure, cardiologist in Japan usually underwent coronary angiography, therefore coronary artery disease more tended to detect. However, after propensity score matched cohort, the difference of rate of coronary artery disease was set off.

#7.Is it expected to have such a low rate of AF in such elderly patients? This is strikingly different for USA and Europe.

Thank you for your valuable comments.

In studies of USA and Europe country regarding TAVI procedure, the prevalence of AF in patients with severe AS who underwent TAVI was about 40%, and this is strikingly different for our data. There was a study that prevalence of atrial fibrillation in the general population of Japan was lower than USA [2]. In addition, another TAVI registry in Japan showed that only 20% of the patients who underwent TAVI had atrial fibrillation [3]. We consider that the prevalence of AF was lower than US data even in such elderly patients.

Reference:

[1] Handa N, Kumamaru H, Torikai K, Kohsaka S, Takayama M, Kobayashi J, et al. Learning Curve for Transcatheter Aortic Valve Implantation Under a Controlled Introduction System- Initial Analysis of a Japanese Nationwide Registry. Circulation journal : official journal of the Japanese Circulation Society. 2018;82(7):1951-8.

[2] Inoue H, Fujiki A, Origasa H, Ogawa S, Okumura K, Kubota I, et al. Prevalence of atrial fibrillation in the general population of Japan: an analysis based on periodic health examination. International journal of cardiology. 2009;137(2):102-7.

[3] Hioki H, Watanabe Y, Kozuma K, Kawashima H, Nagura F, Nakashima M, et al. The MAGGIC risk score predicts mortality in patients undergoing transcatheter aortic valve replacement: sub-analysis of the OCEAN-TAVI registry. Heart and vessels. 2019.

 

<Response to the comments of the reviewer #2>

Reviewer #2:

In this observational retrospective study authors aimed at comparing TAVI to medical treatment using historical cohort data. The study is not timely since the question asked has been previously answered in much more robust studies. However, it could still be interesting regarding the long-term differences between the two groups. But authors should provide a more robust methodological and statistical approach and would benefit from simplification (the sensitivity and subgroup analyses provide little new insight).

#1.Not a timely study, as attested by the use of Sapien XT valves, 37% of alternative access, with probably transapical as first alternative (Results page 22 “trans-femoral approach was selected only in 63% of patients”) with a comparator based on a historical pre-TAVI era cohort. Must be clearly stated that comparison is not applicable to current clinical practice

Thank you very much for your valuable suggestion. As you pointed out, this study compared TAVI with early experience in Japan and conservative therapy in before introduction TAVI. About 37% of patients who underwent TAVI were selected alternative approach and almost all patients were undergone TAVI under general anesthesia. This is not applicable to current TAVI practice ,although we consider that the data of early experience was important to evaluate long-term results over 2 years. 

We added the following statement regarding the difference from current clinical practice in the limitations section.

Limitations (Page 21, Line 9 to Page 21, Line 12)

Fifth, patients who underwent TAVI were early experience data in Japan, therefore about 37% of patients were selected alternative approach and almost all patients underwent TAVI under general anesthesia. This is quite different from current TAVI practice and this could not applicable to current TAVI practice.

#2.Method page 15 “We also excluded those patients … might also be contraindicated for TAVI” Authors should clarify what patients were excluded here and reformulate this statement because TAVI is a clear indication for a large proportion of patients which are contraindicated to SAVR. Please also clarify the flow chart figure 1 with regard to the box “conservative group with formal indication of AVR …”

Thank you for your valuable suggestion. We excluded patients who were considered to be contraindicated aortic valve replacement (TAVI or SAVR) because of malnutrition, muscle weakness, cognitive impairment, and expected poor prognosis due to morbidities other than heart disease.

For detail, 146 patients excluded contraindicated because of malnutrition (61 patients) , muscle weakness (26 patients), cognitive impairment (101 patients), and 46 patients excluded because of expected poor prognosis due to morbidities other than heart disease.

We added this details of exclusion you suggested to method, and modified Fig 1 in "Response to reviewer file" added on the end of "revised file".

Materials and Methods (Page 8, Line 10 to Page 8, Line 12)

We also excluded those patients who were regarded as contraindicated for SAVR by the attending physicians (malnutrition, muscle weakness, cognitive impairment, and expected poor prognosis), because these patients were considered to be contraindicated for TAVI.

The detail of “conservative group with formal indication of AVR” was written below in the same box in Figure 1.

The formal indication of AVR were symptomatic patients, asymptomatic patients with Vmax>=5m/, or asymptomatic patient with LVEF<50% mentioned in AHA/ACC Guideline [1].

We clearly described as below.

Materials and Methods (Page 8, Line 6 to Page 8, Line 9)

we excluded those patients on hemodialysis (HD) in whom TAVI has not been yet approved in Japan, and those asymptomatic patients with Vmax <5m/s and left ventricular ejection fraction (LVEF) >=50%, who are regarded as candidates for watchful waiting according to the guidelines

#3.Statistical analysis page 17 “Because some variables were not well balanced … further adjustment by using the Cox proportional hazard models” In case of PS matching failure please consider removing PS matching step and apply multivariable Cox upfront. 

Thank you for your valuable suggestion. 

In this study, we combined two different studies and the difference of baseline characteristics between two groups was significant. Regarding this problem, we discussed with statistician and decided to use PS matching. The statistical method of using cox proportional hazard model with residual variables to adjust after PS matching have been conducted in prior studies [2][3][4].

To confirm robustness of PS matching method, we also conducted cox proportional hazard models as sensitivity analysis, and the results were consistent with PS matching.

#4.Statistical analysis “We also performed subgroup analyses in terms of age, sex, STS score, LVEF, and high/low gradient AS in the propensity-score matched cohort” As reported by authors, the PS matching failed, therefore building subgroups for this cohort is likely to provide biased results.

Thank you for your valuable suggestion. 

To adjust residual confounders, we also conducted cox proportional hazard model after PS matching for subgroup analyses. As you pointed out, there were not new finding in subgroup analysis. Our aim to conduct subgroup analysis was to confirm consistency of results about primary outcomes.

#5 Please consider separating all-cause mortality and HR rehospitalization all over the manuscript. Those 2 outcomes have little in common and the database is likely able to provide these data.

Thank you very much for your suggestion. In the study, primary outcomes were all-cause mortality and HF hospitalization, but these were not composite endpoint. In our study, these two outcomes described respectively.

#6.Please provide information regarding TAVI procedural characteristics in a Table: valves used, labels/generations, pathways, TEE use, general versus local anaesthesia, complications, etc

Thank you very much for your comment. We described TAVI procedural characteristics in S2 Table below. We show reference table in "Response to reviewer file" added on the end of "revised file".

We described approach site, procedure time, valve size, type of anesthesia, ECMO and complications. However, we did not have data about TEE use.

Reference:

[1] Nishimura RA, Otto CM, Bonow RO, Carabello BA, Erwin JP, 3rd, Fleisher LA, et al. 2017 AHA/ACC Focused Update of the 2014 AHA/ACC Guideline for the Management of Patients With Valvular Heart Disease: A Report of the American College of Cardiology/American Heart Association Task Force on Clinical Practice Guidelines. Journal of the American College of Cardiology. 2017;70(2):252-89.

[2] Taniguchi T, Morimoto T, Shiomi H, Ando K, Kanamori N, Murata K, et al. Initial Surgical Versus Conservative Strategies in Patients With Asymptomatic Severe Aortic Stenosis. Journal of the American College of Cardiology. 2015;66(25):2827-38.

[3] Birkhead JS, Weston CF, Chen R. Determinants and outcomes of coronary angiography after non-ST-segment elevation myocardial infarction. A cohort study of the Myocardial Ischaemia National Audit Project (MINAP). Heart (British Cardiac Society). 2009;95(19):1593-9.

[4] Bajaj JS, Ratliff SM, Heuman DM, Lapane KL. Proton pump inhibitors are associated with a high rate of serious infections in veterans with decompensated cirrhosis. Aliment Pharmacol Ther. 2012;36(9):866-74.

---

## [Decision Letter · Decision Letter 1]

12 Sep 2019

[EXSCINDED]

Transcatheter Aortic Valve Implantation Versus Conservative Management for Severe Aortic Stenosis in Real Clinical Practice

PONE-D-19-17543R1

Dear Dr. Naritatsu Saito,

We are pleased to inform you that your manuscript has been judged scientifically suitable for publication and will be formally accepted for publication once it complies with all outstanding technical requirements.

With kind regards,

Cécile Oury

Academic Editor

PLOS ONE

Additional Editor Comments:

The reviewer 1 and myself considered that the authors have adequately addressed the issues that were raised. The reason why the study is not a timely study according to current clinical practices has been well justified by the authors, missing patient informations have been provided, and statistical analyses have been adapted as requested.

Reviewers' comments:

Reviewer's Responses to Questions

**Comments to the Author**

1. If the authors have adequately addressed your comments raised in a previous round of review and you feel that this manuscript is now acceptable for publication, you may indicate that here to bypass the “Comments to the Author” section, enter your conflict of interest statement in the “Confidential to Editor” section, and submit your "Accept" recommendation.

Reviewer #1: All comments have been addressed

2. Is the manuscript technically sound, and do the data support the conclusions?

Reviewer #1: Yes

3. Has the statistical analysis been performed appropriately and rigorously? 

Reviewer #1: Yes

4. Have the authors made all data underlying the findings in their manuscript fully available?

Reviewer #1: Yes

5. Is the manuscript presented in an intelligible fashion and written in standard English?

Reviewer #1: Yes

6. Review Comments to the Author

Reviewer #1: Thanks for having worked on the manuscript according to the comments we previously did.

The paper is OK and is providing data that are insteresting.

No major comment remains about the form and,

the content is OK.

7. PLOS authors have the option to publish the peer review history of their article (what does this mean?). If published, this will include your full peer review and any attached files.

Reviewer #1: Yes: DONAL Erwan

---

## [Editor Report · Acceptance letter]

18 Sep 2019

PONE-D-19-17543R1 

Transcatheter Aortic Valve Implantation Versus Conservative Management for Severe Aortic Stenosis in Real Clinical Practice 

Dear Dr. Saito:

I am pleased to inform you that your manuscript has been deemed suitable for publication in PLOS ONE. Congratulations! Your manuscript is now with our production department. 

With kind regards,

on behalf of

Dr. Cécile Oury 

Academic Editor

PLOS ONE